# Resonance Raman Spectro-Electrochemistry to Illuminate Photo-Induced Molecular Reaction Pathways

**DOI:** 10.3390/molecules24020245

**Published:** 2019-01-10

**Authors:** Linda Zedler, Sven Krieck, Stephan Kupfer, Benjamin Dietzek

**Affiliations:** 1Department Functional Interfaces, Leibniz Institute of Photonic Technology Jena, Albert-Einstein-Strasse 9, 07745 Jena, Germany; linda.zedler@leibniz-ipht.de; 2Institute of Inorganic and Analytical Chemistry, Friedrich-Schiller-University Jena, Humboldtstrasse 8, 07743 Jena, Germany; sven.krieck@uni-jena.de; 3Institute of Physical Chemistry and Abbe Center of Photonics, Friedrich-Schiller-University Jena, Helmholtzweg 4, 07743 Jena, Germany; stephan.kupfer@uni-jena.de

**Keywords:** UV-vis, resonance Raman, in situ spectro-electrochemistry, TDDFT, ruthenium bis-terpyridine complex

## Abstract

Electron transfer reactions play a key role for artificial solar energy conversion, however, the underlying reaction mechanisms and the interplay with the molecular structure are still poorly understood due to the complexity of the reaction pathways and ultrafast timescales. In order to investigate such light-induced reaction pathways, a new spectroscopic tool has been applied, which combines UV-vis and resonance Raman spectroscopy at multiple excitation wavelengths with electrochemistry in a thin-layer electrochemical cell to study [Ru^II^(tbtpy)_2_]^2+^ (tbtpy = tri-*tert*-butyl-2,2′:6′,2′′-terpyridine) as a model compound for the photo-activated electron donor in structurally related molecular and supramolecular assemblies. The new spectroscopic method substantiates previous suggestions regarding the reduction mechanism of this complex by localizing photo-electrons and identifying structural changes of metastable intermediates along the reaction cascade. This has been realized by monitoring selective enhancement of Raman-active vibrations associated with structural changes upon electronic absorption when tuning the excitation wavelength into new UV-vis absorption bands of intermediate structures. Additional interpretation of shifts in Raman band positions upon reduction with the help of quantum chemical calculations provides a consistent picture of the sequential reduction of the individual terpyridine ligands, i.e., the first reduction results in the monocation [(tbtpy)Ru(tbtpy^•^)]^+^, while the second reduction generates [(tbtpy^•^)Ru(tbtpy^•^)]^0^ of triplet multiplicity. Therefore, the combination of this versatile spectro-electrochemical tool allows us to deepen the fundamental understanding of light-induced charge transfer processes in more relevant and complex systems.

## 1. Introduction

Photo-induced electron transfer cascades represent the key element for artificial solar energy conversion. While there is an urgent need for novel concepts to satisfy the world’s hunger for energy, e.g., dye-sensitized solar cells (DSSCs) or molecular photocatalysts, both the efficiency and stability of artificial photosynthetic systems is still insufficient for large-scale application [1]. This is due to the fact that besides the broad and strong absorption, chemical stability, and appropriate electrochemistry, the photoexcitation and subsequent electron transfer dynamics in intramolecular artificial photosynthetic systems mainly determine their efficiency [2]. To correlate structure, function, and finally efficiency of photoactive systems a complete and systematic spectroscopic investigation of electron transfer processes is indispensable. The underlying photo-induced charge transfer processes occur in the femtosecond to microsecond range, so these processes can only be studied directly using time-resolved techniques. Long-lived intermediates that emerge in the course of the photo-induced reaction play a key role, since competing reaction pathways separate at these metastable intermediates. However, the characterization of intermediate states is challenging due to their high reactivity, still comparably short lifetimes, and low concentrations. In this contribution, a new setup utilizing a custom made thin-layer spectro-electrochemical cell is realized, which combines UV-vis- and resonance Raman (RR) spectro-electrochemistry (SEC) to study electronic and structural changes during one- and multielectron reduction [3,4,5,6]. RR-SEC is suitable to localize the initial photoexcitation process of intermediate structures since exclusively these vibrations are intensified, which are associated with structural changes initiated by electronic absorption. To identify intermediates of artificial light-driven energy conversion assemblies as well as to study their photophysical properties, which substantially impact their function, a redox sensitive resonance Raman setup including a thin-layer cell construction has been designed. This RR-SEC setup was optimized to elucidate molecular structure changes of intermediates mimicked via electrochemical reduction and to unravel the mechanism of electron transfer cascades in solution and under inert conditions. This knowledge forms the basis for synthetic optimization of artificial molecular light-harvesting systems. [Ru(tbtpy)_2_]^2+^ (tbtpy = tri-tert-butyl-2,2′:6′,2′′-terpyridine) is used as a model compound for the photo-activated electron donor in structurally related molecular and supramolecular systems [4,7,8,9,10,11,12] to study the fundamental photo-induced electron transfer processes on a molecular level.

The importance of tpy ligand containing coordination compounds arises from planar and stiff coordination of these ligands—leading to the formation of the *mer*-[Ru^II^(tpy)_2_]^2+^ [13,14,15]. Furthermore, functionalization of the tpy ligands with electron-donating and/or withdrawing groups [13,16,17,18,19,20,21,22] allows a fine tuning of the redox and electronic properties of its Ru^II^ compounds [23] with potential application as photosensitizer ranging from organic light emitting diodes (OLEDs) [15,24,25] and photocatalysis [4,15], to dye-sensitized solar cells (DSSCs) [15,26,27,28,29]. The high stability combined with very favorable redox properties render [Ru(tpy)_2_]^2+^ complexes ideal model systems to study the photo-induced electron transfer processes of short-lived intermediates mimicked by its singly and doubly reduced species.

Previously reported UV-vis-SEC studies on singly reduced species of [Ru(tpy)_2_]^2+^ showed that the excess charge populates a π* orbital of the tpy ligand sphere [30,31,32]. The formation of [(tpy)Ru(tpy)^•−^]^+^ is also in agreement with low-temperature electron spin resonance (ESR) studies performed on singly reduced products of [Fe(tpy)_2_]^2+^ and [Ru(tpy)_2_]^2+^ complexes [30,31,33], as well as for [Fe(bpy)_3_]^2+^ and [Ru(bpy)_3_]^2+^ [31], where exclusively the doublet species (S = 1/2) was observed. Low-temperature ESR and UV-vis-SEC experiments upon double reduction suggest a sequential reduction of the individual tpy ligands, i.e., the formation of [(tpy^•−^)Ru(tpy^•−^)] [14,31,32,33]. The redox chemistry of terpyridine-based ruthenium complexes is commonly assessed by cyclic voltammetry and ESR as well as by UV-vis/NIR spectro-electrochemistry assisted by computational methods [14,32,34]. Despite the fact that these methods are highly indicative in elucidating the formation of the aforementioned reduction products of various functionalized [Ru(tpy)_2_]^2+^ complexes, insight regarding the underlying structural changes—in particular on fast excited state dynamics in the reduced complexes—is still unavailable. Redox sensitive RR spectroscopy could help to fill this gap, since this method can probe the vibrational changes of individual chromophores by choosing the excitation wavelength close to a specific absorption band of the electrochemically generated species. Therefore, the present joint spectroscopic–theoretical work provides a comprehensive study of the sequential electrochemical reduction of the [Ru(tbtpy)_2_]^2+^ photosensitizer based on experimental in situ UV-vis and RR-SEC corroborated by quantum chemical simulations performed at the density functional (DFT) and time-depended DFT (TDDFT) levels of theory. This approach provides to the best of our knowledge the first and convincing experimental evidence for the previously proposed reduction mechanism by direct spectroscopic characterization of the structure of the singly and doubly reduced form of [Ru(tbtpy)_2_]^2+^.

## 2. Results and Discussion

### 2.1. UV-Vis Spectro-Electrochemistry

The investigated ruthenium complex, [Ru(tbtpy)_2_]^2+^, depicted in Figure 1A (inset), contains two identical tri-*tert*-butyl-2,2′:6′,2′′-terpyridine (tbtpy) ligands coordinated to a Ru^II^ center. The complexes reductive electrochemistry is based on reversible, one-electron ligand-centered processes [14,30,35]. In the upper right inset of Figure 1A results from voltammetric measurements of the first and second reduction process of [Ru(tbtpy)_2_]^2+^ are displayed. The data were acquired in a spectro-electrochemical 1 mm thin-layer cell equipped with a platinum grid working electrode. The voltammogram in ACN/0.1 M TBABF_4_ [Ru(tbtpy)_2_]^2+^ shows two reversible one-electron reduction steps. [Ru(tbtpy)_2_]^2+^ is singly and doubly reduced at E_1/2_ = −1.36 V and E_1/2_ = −1.61 V vs. Ag/AgCl pseudo-reference electrode (E_1/2_ = −1.78 V/−2.03 V vs. Fc^+^/Fc), respectively.

The experimental UV-vis absorption spectrum of non-reduced [Ru(tbtpy)_2_]^2+^ in ACN (Figure 1A) exhibits one broad absorption band in the visible region centered at 481 nm (2.58 eV) and one broad band in the UV with a maximum at 309 nm (4.01 eV), which agrees well with the calculated UV-vis absorption spectrum (Figure 2A) [14,16,19,30,36]. TDDFT associates the broad absorption band in the visible range with a superposition of eight electronic states of metal-to-ligand charge transfer (MLCT) character, namely S_1_, S_2_, S_5_, S_7_, S_8_, S_9_, S_10_ and S_11_ (Figure 2A), which are pairwise degenerate due to the molecular symmetry of [Ru(tbtpy)_2_]^2+^. The high-energy absorption at 308 nm is dominated by two degenerated bright intraligand (IL) states (S_27_ and S_28_) at 4.09 eV. Detailed information concerning the individual excited states contributing to the absorption spectrum of [Ru(tbtpy)_2_]^2+^ and MOs involved in the leading transitions is collected in Appendix A. In general, a blue shift of approximately 0.3 eV is observed for the MLCT states contributing to the MLCT band in comparison to the experimental band position; such deviation is typical for MLCT states at the TDDFT level of theory using the present computational protocol [37,38].

Electronic absorption spectra of [Ru(tbtpy)_2_]^2+^ were measured at the first and second reduction potential (Figure 1A, red and blue spectra). For a better visualization of the spectral changes during the two reduction steps, difference spectra were calculated (Figure 1B). The absorption spectrum recorded during cycling a slow voltammogram of [Ru(tbtpy)_2_]^2+^ in the potential range of −1.20 and −1.50 V (scan rate of 5 mVs^−1^) is shown in Figure 1A. The decrease of the MLCT absorption band at 481 nm is accompanied by the growth of new absorption features between 334 and 434 nm as well as between 490 and 700 nm peaking at 355, 538, and 630 nm (Figure 1 and Figure 2B, red graph). The new band at 630 nm has been assigned to a ligand centered transition of the one electron reduced radical form of the tpy ligand [14,30]. In agreement with previous studies, a very weak and broad absorption band in the NIR is observed upon reduction [13,39]. The observed spectral changes increase with increasing reduction potential—in particular, the shoulders at 414 and 630 nm and the broad band around 800 nm become significantly more pronounced during the second reduction in comparison to the first reduction of [Ru(tbtpy)_2_]^2+^ (Figure 1).

Experimental and theoretical spectra of singly and doubly reduced [Ru(tbtpy)_2_]^2+^ are in good agreement (Figure 2). The electrochemical one-electron reduction of [Ru(tbtpy)_2_]^2+^ leads to a population of the π* orbital of one tbtpy ligand forming the coordinated radical anion (tbpy^•^)^−^ in [(tbtpy)Ru(tbtpy^•^)]^+^ [30,40]. Therefore, the unpaired electron is localized on a single tbtpy ligand and the corresponding monocation [(tbtpy)Ru(tbtpy^•^)]^+^ is present in the doublet multiplicity (S = 1/2) as can be concluded from the calculation (Appendix A). In case of the second reduction process two different scenarios have to be taken in account: (i) Both electrons occupy one single antibonding molecular orbital (centered at one tbtpy ligand [(tbtpy)Ru(tbtpy^••^)]^0^), which leads to the formation of a doubly reduced singlet species (S = 0), and (ii) each of the two electrons involved in the reduction process is located on one of the terpyridine ligands [(tbtpy^•^)Ru(tbtpy^•^)]^0^. The latter scenario consequently leads to a doubly reduced triplet species (S = 1). However, the obtained electronic ground state energies for the doubly reduced singlet (−4.84 eV using restricted DFT) and triplet (−5.17 eV using unrestricted DFT) species indicate that the triplet state is energetically favored.

To assign the spectral changes specifically to the singly and doubly reduced complex, a normalized absorbance Abs_norm_ = Abs(E_WE_) − Abs(ocp)/Abs(1st reduction) has been calculated and plotted at 355, 414, 426, 536, 630, and 800 nm as a function of the reductive potential (Figure 3). First, by subtraction of the absorbance at ocp, the difference spectrum is calculated focusing on spectral changes. By normalization to the absorbance at the first reduction potential, Abs_norm_ measures the concentration of the singly reduced complex according to Beer-Lambert’s law. Hence, within the first reduction potential range, Abs_norm_ resembles the CV and the formation of the singly reduced complex and is identical, i.e., non-dispersive, for all wavelengths (Figure 3A). When increasing the reductive potential further, Abs_norm_ becomes dispersive (Figure 3B). Large changes in Abs_norm_ indicate strong absorption of the doubly reduced complex, particularly at 414, 426, and 800 nm. That proves, that by increasing reductive potential (i) the concentration of singly reduced complex increases and additionally that (ii) the doubly reduced form is generated, which is characterized by absorption features at 414, 426, and 800 nm.

The TDDFT simulations on the singly reduced complex allowed us to assign the main spectral alterations between 800 and 550 nm to low-lying intra ligand (IL) states of π*_tbtpy_π*_tbtpy_ nature (see Appendix A for more details). Unfortunately, the comparison of results obtained from in situ UV-vis-SEC and TDDFT simulations for the doubly reduced system neither favor one of the above-mentioned possible reduction processes, i.e., singlet or triplet. Furthermore, the sequential charge transfer processes upon photoexcitation in doubly reduced [Ru(tbtpy)_2_]^2+^ as well as the underlying changes in the electron density distribution cannot be elucidated since no information about structural alterations and the fast-excited state dynamics accompanying electrochemical reduction can be extracted. Therefore, electrochemical potential controlled RR spectroscopy is applied to localize the excess charges and to probe the vibrational properties of the reduced ligand chromophores in singly and doubly reduced [Ru(tbtpy)_2_]^2+^. This technique allows us to monitor changes in the electron-density distribution upon photoexcitation since selectively those modes are enhanced which are affected by the structural reorganization.

### 2.2. Resonance Raman Spectro-Electrochemistry

In the following section the results from RR spectroscopic studies on the non-reduced and the electrochemically reduced [Ru(tbtpy)_2_]^2+^ complex will be explained and subsequently interpreted, assisted by quantum chemical calculations. The laser line at 514 nm is resonant with both the non-reduced (MLCT transitions) and the reduced (IL transitions) forms of [Ru(tbtpy)_2_]^2+^ (Figure 1). However, since the resonance Raman enhancement factors of the reduced complexes are not known, it is not possible to quantitatively determine the concentrations of the reduced complex in the laser focus. Furthermore, due to incomplete reduction of the solution, the experiment will principally detect a mixture of non-reduced and reduced [Ru(tbtpy)_2_]^2+^ within the laser focus (Figure 4B–D), while the calculated spectra are of the pure complexes in their respective redox state only. Figure 4 displays the RR-SEC spectra of the non-reduced (A), the singly reduced (*E*_pol_ = −1.45 V) (B), and the doubly reduced species (*E*_pol_ = −1.80 V) (C and D) together with the respective calculated RR spectra for each reduction state.

The RR spectrum of the non-reduced complex is characterized by intense bands centered at 1612, 1535, 1479, 1449, 1326, 1294, 1278, 1250, and 1205 cm^−1^ (Figure 4A). TDDFT predicts eight (pairwise degenerate) MLCT transition, namely into S_1_, S_2_, S_5_, S_7_, S_8_, S_9_, S_10_, and S_11_, in resonance with the laser at 515 nm (Figure 2A). Detailed information concerning the individual excited states contributing to the absorption spectrum of [Ru(tbtpy)_2_]^2+^ and MOs involved in the leading transitions is collected in Appendix A. The simulated RR spectra, taking into account contributions of these eight MLCT states, allowed the assignment of the measured Raman features to specific vibrational modes of the terpyridine ligands (modes 187, 183, 177, 175, 163, 148, 144, 142, and 131, see Appendix A). An excellent agreement between experiment and theory is achieved for both, the band positions with a mean absolute deviation of merely 8 cm^−1^ and for the intensity pattern (Figure 4A). The RR spectrum of the electrochemically reduced [Ru(tbtpy)_2_]^2+^ differs from the spectrum of the non-reduced species by band shape, spectral position, and intensity. The intensity of the bands at 1612 and 1534 cm^−1^ increases, the bands become broadened and shift slightly to lower wavenumbers upon single reduction of the complex (1610 and 1533 cm^−1^) (Figure 4B). This bathochromic shift is attributed to a reduced force constant and, hence, a lower vibrational energy of the tbtpy modes upon adding an electron into an anti-binding terpyridine orbital. Since the respective π*_tpy_ orbital is clearly localized on one terpyridine ligand, the molecular symmetry is reduced, and the pairwise excited state degeneracy is broken. Thus, the quantum chemical simulations reveal both low-lying π*_tbtpy_π*_tbtpy_ IL and ligand-to-ligand charge transfer (LLCT) transitions originating from the reduced ligand, i.e., D_7_, D_13_, D_14_, and partially D_22_ between 756 (1.64 eV) and 488 nm (2.54 eV), as well as MLCT transitions mainly to the non-reduced ligand, i.e., D_25_, D_35_ and D_57_ at 463, 405, and 345 nm (2.68, 3.06, and 3.60 eV), respectively. These IL, LLCT, and MLCT states were considered to assess the RR signature of the singly reduced doublet species, [(tbtpy)Ru(tbtpy^•^)]^+^, at the given excitation wavelength. Thus, the spectral changes observed at 1610 cm^−1^ can be mainly rationalized by the intense mode at 1605 cm^−1^ (mode 188), which is slightly shifted to lower frequencies with respect to the equivalent mode centered at 1609 cm^−1^ (mode 187) of the non-reduced complex. The changes in the RR intensity pattern around 1533 cm^−1^ are related to a superposition of Raman modes of both the non-reduced (mode 183) and singly reduced species (modes 183, 184, and 196), while a new band is observed at 1464 cm^−1^ (modes 163, 164, 175, 173, 179, and 180). The most prominent active vibrations of the non-reduced species (S = 0) and singly-reduced doublet species are presented in the Appendix A, respectively.

RR spectra recorded upon double reduction of [Ru(tbtpy)_2_]^2+^ closely resemble the spectra of the non-reduced and singly reduced complex (Figure 4C) indicating only minor changes of the ligand geometry upon photoexcitation of the doubly reduced complex. Nonetheless, double reduction causes similar but more pronounced changes of the observed Raman modes at 1534 and 1612 cm^−1^ with respect to the Raman spectrum of the singly reduced complex. These two bands are shifted by 5 and 3 cm^−1^ to lower wavenumbers, respectively, which is resolvable by the spectrograph providing 2 cm^−1^ resolution (750 mm focal length, 1800/mm grating, 20 µm pixel). As indicated in Section 2.1 the performed TDDFT simulations do not allow an unambiguous assignment of the UV-vis-SEC spectrum either to the doubly reduced singlet or triplet species. The triplet species features two singly reduced tpy ligands, and thus low-lying ^3^IL (π*_tbtpy_π*_tbtpy_) are found between 785 and 584 nm (between 1.58 and 2.12 eV), namely T_9_, T_10_, and T_15_, and higher lying MLCT states T_16_, T_32_, T_33_, and T_35_ at 581, 484, 481, 476 nm (2.13, 2.56, 2.57, 2.61 eV). On the other hand, the doubly reduced form features a low-lying π*_tpy_→π*_tpy_ IL state, S_7_ at 726 nm, and a superposition of several medium bright singlet MLCT states (S_11_, S_13_, S_14_ S_16_ S_18_, and S_20_), see Appendix A. However, the comparison of the experimental RR-SEC with the simulated RR spectra for the doubly reduced species of singlet and triplet multiplicity indicates that the second reduction generates the triplet species [(tbtpy^•^)Ru(tbtpy^•^)]^0^ (Figure 4C,D). The increase of the Raman feature at 1609 cm^−1^ as well as the observed bathochromic shift upon the second reduction cannot be correlated to the formation of an intense Raman active mode of the singlet form; however, for the triplet form such an intense mode (187) exists. This mode is equivalent to vibrational modes 187 and 188 of the non-reduced and the singly reduced species at slightly higher frequency, respectively, and thus the Raman band at 1609 cm^−1^ is a marker band for the doubly reduced species of triplet multiplicity. These results discussed confirm the suspected sequential one-electron reductions of [Ru(tbtpy)_2_]^2+^ to its (doubly) reduced analogue [(tbtpy^•^)Ru(tbtpy^•^)]^0^ with triplet character, which is energetically slightly favored by approximately 0.3 eV with respect to the closed-shell singlet, as predicted by DFT calculations [14,30,32,33,41,42]. Therefore, the combination of electrochemistry with UV-vis and RR spectroscopy presents a powerful and unique tool to monitor the formation of radical anions and to study both their structural properties and electronic configuration.

## 3. Methods

### 3.1. Experimental

[Ru(tbtpy)_2_](PF_6_)_2_ was prepared according to the synthesis procedure published elsewhere [37]. All solutions were prepared using acetonitrile (ACN, HPLC-grade, Aldrich, St. Louis, MO, USA) dried over calcium hydride and distilled twice. Dried tetrabutylammonium tetrafluoroborate (TBABF_4_) was used as electrolyte. UV-vis-NIR- and RR-SEC measurements were performed under argon atmosphere using a three-electrode thin-layer spectro-electrochemical cell (Bioanalytical Systems, Inc, West Lafayette, IN, USA). A platinum-gauze (dimension: 6 mm × 7 mm, 80 mesh) and a platinum wire were used as a working and counter electrode, respectively. A chloridized Ag/AgCl electrode (Science Products GmbH, Hofheim, Hessen, Germany) was used as a reference electrode. The electrochemical measurements were performed using an Autolab PGSTAT potentiostat (Filderstadt, Baden-Württemberg, Germany).

The in-situ UV-vis-SEC measurements were performed at ambient temperature using a double UV-vis-NIR spectrometers beam (Cary 5000 UV/Vis spectrometer, Agilent, Santa Clara, CA, USA) with 1 nm spectral resolution and a Fiber Optic AvaSpec ULS2048XL spectrometer (Avantes, Apeldoorn, Veluwe, The Netherlands) with a charged coupled device (CCD, SPEC-10, Roper Scientific, Acton, MA, USA) detector in the wavelength range between 200 and 1100 nm. The RR spectra were excited by the visible lines of an Argon ion laser (Coherent, Innova 300C, Santa Clara, CA, USA) and recorded with an Acton SpectraPro 2758i spectrometer (entrance slit width 100 µm, focal length 750 mm, grating 1800/mm, Acton, MA, USA). A general scheme of the RR-setup is depicted in Figure 5. Measurements were performed in transmission. The Ar-ion laser was spectrally cleaned from plasma lines by a filter (2) before being focused into a 1 mm path length SEC cell (Figure 5B). The cell was customized for oxygen-free operation (Figure 5C). It was filled under inert conditions in a glove box and sealed with a Teflon cap. The cell was then placed in the beam path and the laser focused onto the Pt-grid working electrode by a 10× microscope objective (Plan-Achromat, Olympus, Japan) (3), while the potential at the working electrode was controlled by a potentiostat (5). The scattered light was collimated by a lens (6), and filtered from lasers and Rayleigh-scattered light by a long pass filter (7) before being focused onto a linear collection fiber bundle and transmitted to the spectrograph (9), while the data was subsequently analyzed on a computer.

The sequence of a complete SEC measurement is shown in Figure 5D. First, a CV was recorded either directly in the thin-layer SEC cell or in a conventional electrochemistry cell to determine the redox potentials of the analyte. Subsequently the analyte was investigated by UV-vis SEC. The UV-vis spectrum was first recorded at the open circuit potential and then either during cycling a slow (about 5 mV/s) CV in the reduction potential range or at a suitable constant reduction potential (chronoamperometry). To check the reversibility and to determine possible decomposition a UV-vis spectrum was recorded after the analyte’s reoxidation. For the RR SEC the laser excitation wavelength was chosen within a reduction induced absorption band (Figure 5D, middle). The resonance Raman spectra were measured analogous to the UV-vis measurements. The RR-SEC spectra were collected during the electrode polarizing at the first and second reduction potential for 200 s. To adjust the experimental setup and check for possible photochemical degradation processes, RR spectra were collected at the open-circuit potential (ocp) before and after spectro-electrochemical data acquisition (Figure 5D, right). In total 10 spectra were accumulated using an integration time of 20 s. The RR-SEC spectra of the non-reduced, singly, and doubly reduced species were collected between 1150 and 1650 cm^−1^ upon excitation at 515 nm. A laser power of 37 mW was used. The Raman signals were detected using a liquid-nitrogen cooled CCD (SPEC-10, Roper Scientific, Trenton, NJ, USA). The recorded RR-SEC spectra were background corrected and normalized to the ACN band at 1373 cm^−1^ before subtracting the solvent spectrum.

For the interpretation of the experimental data quantum chemical calculations were carried out using density functional (DFT) and time-depended DFT (TDDFT) levels of theory.

### 3.2. Computational Details

To reduce the computational cost of the simulations without affecting the spectroscopic properties of the [Ru(tbtpy)_2_]^2+^ complex, the six *tert*-butyl groups were approximated in the calculations by methyl groups. The structural and electronic data for the complex (see Appendix A) within its non-reduced (singlet), the singly reduced (doublet), and the doubly reduced forms (singlet and triplet) were obtained from quantum chemical calculations performed with the GAUSSIAN 09 program (Wallingford, CT, USA) [43]. The geometries, vibrational frequencies, and normal coordinates of the electronic ground state were calculated by means of DFT with the XC functional B3LYP [44,45]. This functional provides a balanced description of the absorption features of ruthenium polypyridyle complexes [46,47]. The 6-31G(d) double-ζ basis set [48] was employed for all main group elements. This approach has been shown to be adequate for the calculation of Raman spectra [3,5,37]. The 28-electron relativistic core potential MWB [48] was applied with its basis set for the ruthenium atom, that is, 4s, 4p, 4d, and 5s electrons were treated explicitly, while the first three inner shells were described by the core pseudo-potential. To correct for the lack of anharmonicity and the approximate treatment of electron correlation the harmonic frequencies were scaled by the factor 0.97 [49]. Vertical excitation energies, oscillator strengths, and analytical Cartesian energy derivatives of the excited states were obtained from TDDFT calculations within the adiabatic approximation with the same XC functional, basis set, and pseudo-potential.

The absorption spectra were simulated by means of the excitation energies and oscillator strengths for the 80 lowest excited states of the respective ground state multiplicity. The effects of the interaction with a solvent (acetonitrile, ε = 35.688, n = 1.344) on the geometry, vibrational frequencies, excitation energies, and excited state gradients were taken into account by the integral equation formalism of the polarizable continuum model [50]. The non-equilibrium procedure of solvation was used for calculating the excitation energies and the excited state gradients, this method is well adapted for processes, where only the fast reorganization of the electronic distribution of the solvent is important.

The RR spectra were calculated within the independent mode displaced harmonic oscillator model (IMDHOM). Detailed information with respect to the simulation of RR intensities within the IMDHOM can be found in References [51] and [52] as well as in the references therein. RR intensities have been calculated for the non-reduced complex using the S_1_, S_2_, S_5_, S_7_, S_8_, S_9_, S_10_, and S_11_ states, whereas the S_1_, S_2_, and S_5_ states have been red-shifted by 2500 cm^−1^. For the singly reduced system the D_13_, D_20_, and D_25_ states have been used, while the excitation energy of all three states were bathochromically shifted by 1500 cm^−1^. In order to simulate the RR intensity pattern of the doubly reduced species, the calculations were performed within singlet and triplet multiplicity. Thus, in case of the doubly reduced singlet the states S_11_, S_13_, S_14_, S_16_, S_18_, S_20_, S_25_, and S_26_ were used, while no correction to the vertical excitation energies was applied. The RR intensity pattern within triplet multiplicity was obtained taking into account contributions from the states T_15_, T_16_, T_32_, T_33_, and T_35_; all respective excitation energies were red-shifted by 700 cm^−1^.

## 4. Conclusions

With the assistance of computational studies, the combination of UV-vis- and RR-SEC provides profound insights (i) into structural changes accompanying electrochemical reduction, (ii) the photophysical properties of photo-excited states of radical anions of photo-sensitizers, and (iii) the electronic configuration of these complexes. All these properties severely impact the function of molecular systems for artificial light-driven energy conversion and need to be tailored to improve the stability and efficiency of these structures. In this regard a redox sensitive resonance Raman setup including a thin-layer cell construction has been designed and optimized. The comparison of TDDFT calculations and in situ RR spectro-electrochemical studies on the model compound [Ru(tbtpy)_2_]^2+^ (tbtpy = tri-*tert*-butyl-2,2′:6′,2′′-terpyridine) provides a first and convincing experimental evidence for substantiating the previously proposed reduction mechanism [14,30,32,41,42]. The first reduction leads to a coordinated radical anion yielding the corresponding monocation [(tbtpy)Ru(tbtpy^•^)]^+^. Electronic spectra obtained during the first ligand-centered reduction display multiple absorption bands in the visible region which have been assigned by TDDFT calculations to IL (π*_tpy_→π*_tpy_) and MLCT transitions. For the doubly reduced complex, two spin states, namely singlet and triplet, are generally conceivable, which cannot be clearly distinguished by interpreting the UV-vis-SEC results only. The comparison of Resonance Raman spectra of the doubly reduced complex—as obtained by RR-SEC and TDDFT—proves that the ground state of the doubly reduced complex is of triplet multiplicity. The two electrochemically introduced electrons are distributed over both terpyridine ligand, i.e., [(tbtpy^•^)Ru(tbtpy^•^)]^0^ is formed. Ligand-based reduction leads to a decrease of the binding strength within the terpyridine ligands, which is reflected by shifts to a lower frequency of associated RR bands. The herein applied spectro-electrochemical tools are well suited to analyze the reduction processes in molecular systems and to investigate structural changes during light-induced multi-electron charge transfer processes. This knowledge will help to address mechanistic considerations and to optimize the structure of individual components used for light-driven solar energy conversion.

## Figures and Tables

**Figure 1 molecules-24-00245-f001:**
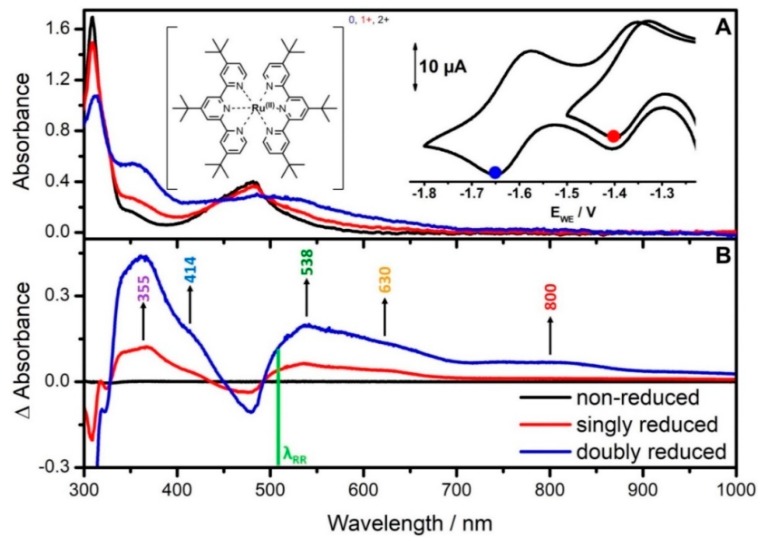
In situ (**A**) UV-vis-NIR absorption and (**B**) difference spectra of [Ru(tbtpy)_2_](PF_6_)_2_ collected during the first (red) and the second (blue) reduction wave. Insets in A: Molecular structure and CV of [Ru(tbtpy)_2_](PF_6_)_2_ in ACN containing the 0.1 M TBABF_4_ electrolyte, recorded in the spectro-electrochemistry (SEC) cell. The electrode potential for acquisition of the absorption spectra is marked in the CV (scan rate 5 mV/s, Pt-gauze working, Pt-counter, and Ag/AgCl-pseudo-reference electrodes).

**Figure 2 molecules-24-00245-f002:**
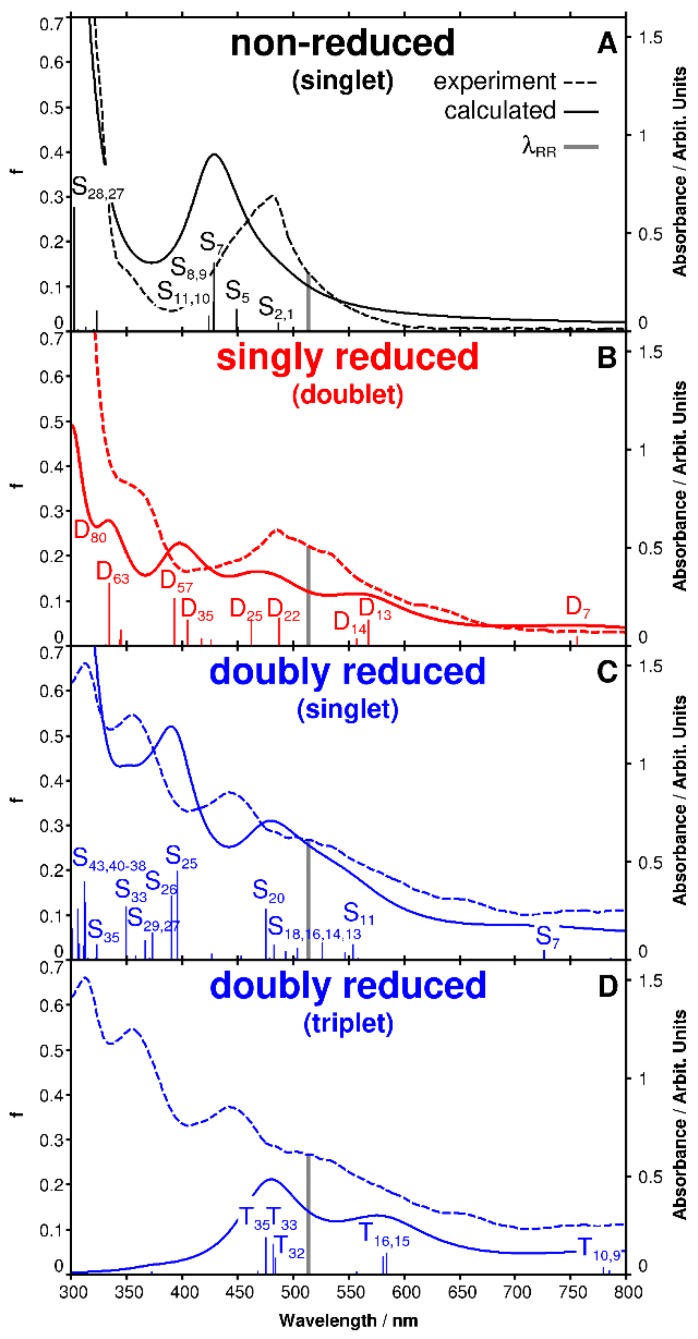
Experimental (dashed line) and theoretical absorption spectra of [Ru(tbtpy)_2_](PF_6_)_2_ (solid line and bars), obtained at the time-depended density functional (TDDFT) level of theory with the B3LYP functional and the 6-31G(d) basis set and applying a PCM model to consider effects of solvation (ACN) of the (**A**) non-reduced specie (E_ocp_), (**B**) singly reduced specie, and (**C**) doubly reduced species of singlet and (**D**) triplet multiplicity, respectively. The vertical gray line indicates the excitation wavelength for the resonance Raman spectro-electrochemistry (RR-SEC) measurements. The experimental spectra for the singly and doubly reduced [Ru(tbtpy)_2_](PF_6_)_2_ (0.55 M in ACN containing 0.1 M TBABF_4_) are collected after polarizing the electrode for 300 s at −1.45 and −1.70 V, respectively.

**Figure 3 molecules-24-00245-f003:**
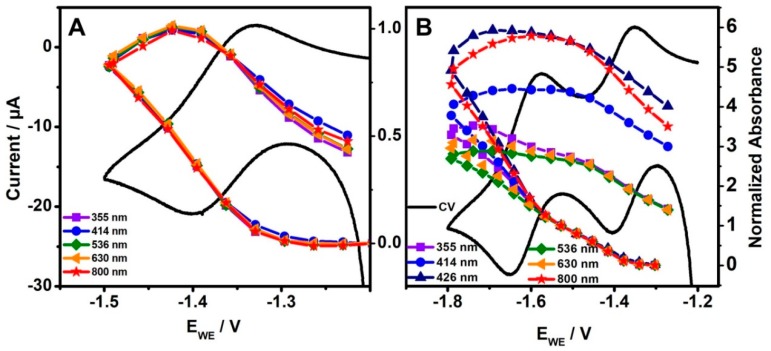
Dependence of the normalized absorbance intensity (normalized to the first reduction) at selected wavelengths on the applied electrode potential for the (**A**) first and the (**B**) second reduction wave.

**Figure 4 molecules-24-00245-f004:**
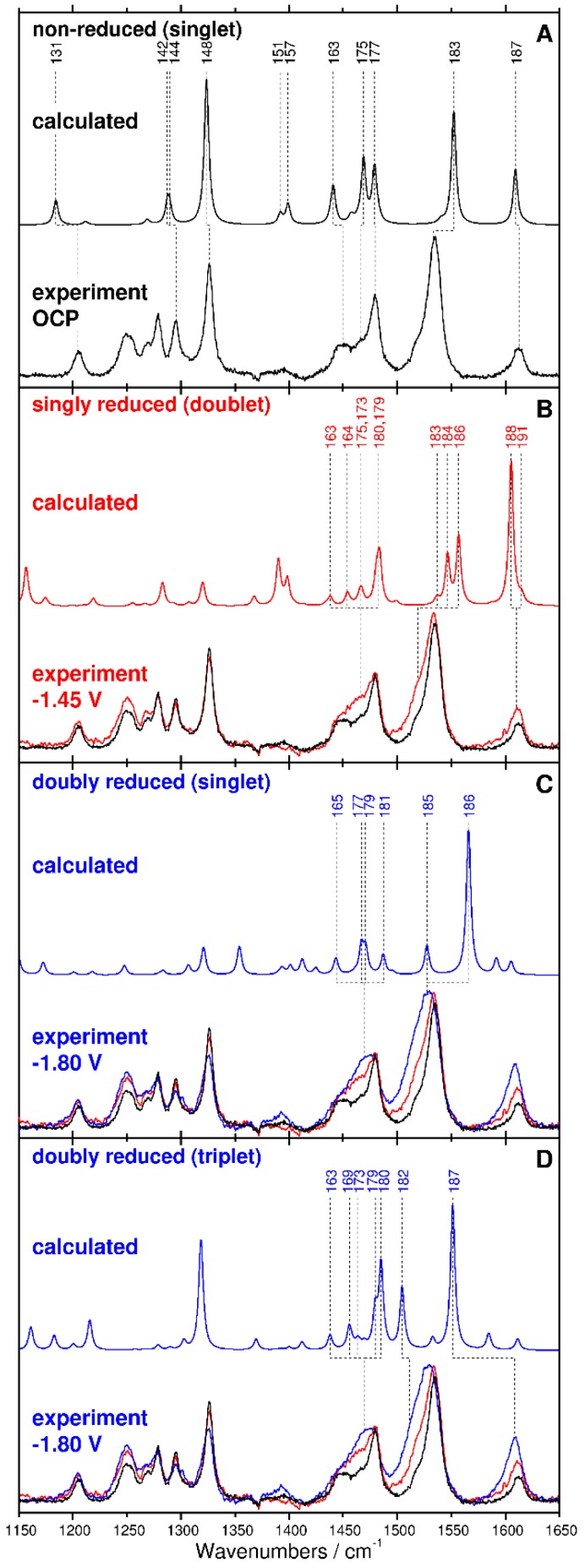
Experimental and calculated RR spectra of non-reduced [Ru(tbtpy)_2_](PF_6_)_2_ collected at (**A**) ocp and non-reduced singlet, at (**B**) −1.45 V and spectrum of the singly reduced doublet species, (**C**) spectrum collected at E_WE_ = −1.80 V and spectrum of doubly reduced singlet species, and (**D**) spectrum collected at E_WE_ = −1.80 V and spectrum of doubly reduced triplet species, excited at 514 nm.

**Figure 5 molecules-24-00245-f005:**
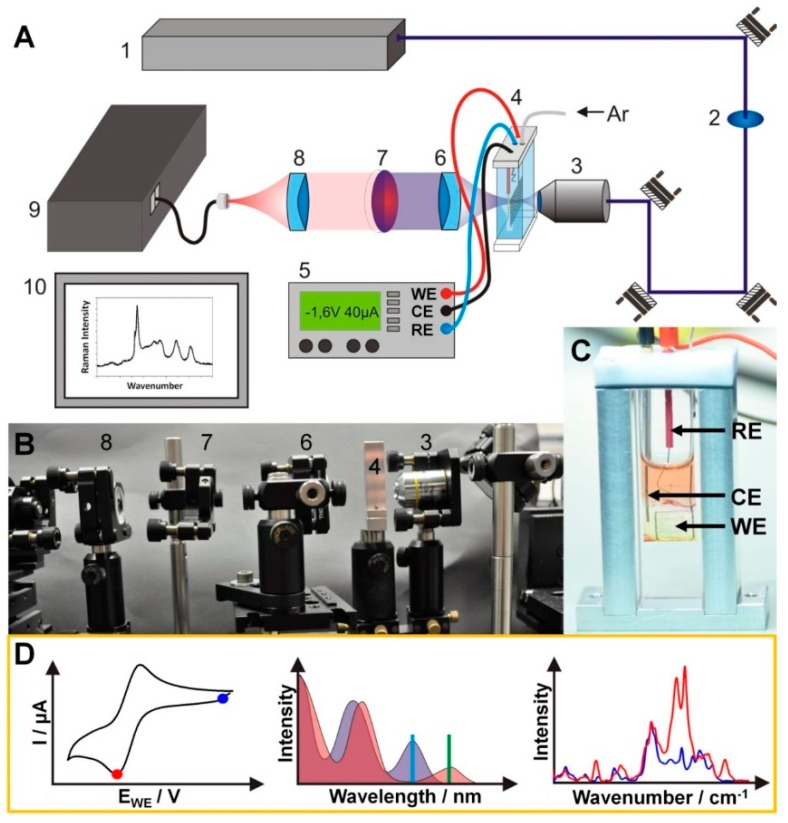
(**A**) Schematic representation and (**B**) a photograph of the measuring setup including an Argon ion Laser (1), laser line bandpass filter (2), a microscope objective (3), the SEC cell (4), a potentiostate (5), UV-vis achromatic optics (6,8), longpass filter (7), spectrometer (9), and a computer (10). (**C**) Photograph of the thin-layer SEC-cell within the custom-made holder, RE: Ag/AgCl reference electrode, CE: Platinum counter electrode, WE: Platinum networking electrode. (**D**) The principle of a SEC measurement starts with recording a CV (**D**, **left**) either directly in the thin-layer SEC cell or in a conventional electrochemistry cell to determine the redox potential of the analyte. The applied potential for the acquisition of UV-vis and Raman spectra are marked with a blue or a red point in the CV. Subsequently the analyte is investigated by UV-Vis- (**D**, **middle**) and resonance Raman (**D**, **right**) spectro-electrochemistry (the blue spectra are recorded at open circuit potential, the red spectra are recorded at a certain redox potential). RR excitation wavelengths are displayed as vertical lines in the UV-vis spectra (**D**, **middle**).

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
