# Peer review of "Resonance Raman Spectro-Electrochemistry to Illuminate Photo-Induced Molecular Reaction Pathways"

_molecules, 2019, doi:10.3390/molecules24020245_

Round 1

Reviewer 1 Report

The manuscript "Resonance Raman Spectro-Electrochemistry to Illuminate Photo-Induced Molecular Reaction Pathways", written by Zedleret al. describes an interesting approach, which enables to localize photo-electrons and to study changes of reactants (mainly structural) and metastable intermediates along the reaction cascades. The manuscript is well written, and all results are adequately described. I have only one minor comment:

1. The Figure 4, describing experimental and calculated resonant Raman spectra should be more discussed in the appropriate section in the manuscript. Differences between calculated and experimental data are described, however, only superficially.

Author Response

Revision of manuscript “Resonance Raman Spectro-Electrochemistry to Illuminate Photo-Induced Molecular Reaction Pathways”

Dear Reviewer 1,

herewith we would like to submit the revised version of the manuscript “Resonance Raman Spectro-Electrochemistry to Illuminate Photo-Induced Molecular Reaction Pathways” to the special issue " Raman Spectroscopy – A Swiss Army Knife ".

Both referees recommended publication with minor revisions. We thank the referees for their critical comments, which we considered in revising the manuscript at hand. For the ease of comparison, we highlighted the major changes using red font in the revised manuscript.

Referee 1 wrote:

The Figure 4, describing experimental and calculated resonant Raman spectra should be more discussed in the appropriate section in the manuscript. Differences between calculated and experimental data are described, however, only superficially.

We agree with the referee, however, while an excellent agreement between experiment and theory is achieved for the parent complex [Ru(tbtpy)2]2+ (Figure 4A) a direct comparison between experimental and calculated resonance Raman spectra of the electrochemically reduced species of [Ru(tbtpy)2]2+ is difficult for several reasons: First, as discussed in the manuscript we do not observe the pure reduced complex upon reduction, but a mixture of reduced and nonreduced [Ru(tbtpy)2]2+ is present in the laser focus and the rate of conversion is unknown. For the doubly reduced complex we assume, that nonreduced, singly reduced and doubly reduced complex are present in the laser focus. Second, the resonance Raman enhancement will be different for nonreduced and the reduced complexes, such that there is no direct link between concentration and Raman band intensity, which is why quantification is not possible. Therefore, only differences between nonreduced and the individual reduced states can be discussed and interpreted. We have added the following paragraph to the manuscript explaining the difficulties of quantifying the experimental results for disentangling the spectral contributions of the nonreduced and reduced complexes. In particular, we state, that the experimental spectra of the reduced complex represent mixtures of nonreduced and reduced complex, while the calculated spectra are of the respective form of the reduced complex only. We now write in section 2.2. Resonance Raman Spectro-Electrochemistry, page 6, line 196:

The laser line at 514 nm is resonant with both the non-reduced (MLCT transitions) and the reduced (IL transitions) forms of [Ru(tbtpy)2]2+ (Figure 1). However, since the resonance Raman enhancement factors of the reduced complexes are not known, it is not possible to quantitatively determine the concentrations of the reduced complex in the laser focus. Furthermore, due to incomplete reduction of the solution, the experiment will principally detect a mixture of non-reduced and reduced [Ru(tbtpy)2]2+ within the laser focus (Figure 4 B - D), while the calculated spectra are of the pure complexes in their respective redox state only. Figure 4 displays the RR-SEC spectra of the non-reduced (A), the singly reduced (Epol=-1.45 V) (B), and the doubly reduced species (Epol=-1.80 V) (C and D) together with the respective calculated RR spectra for each reduction state.

We thank both reviewers for their helpful comments.

With kind regards also on behalf of Prof. Benjamin Dietzek and our co-authors,

Dr. Linda Zedler

Reviewer 2 Report

The authors report an interesting study of the sequential electrochemical reduction of the [Ru(tbtpy)2]2+ photosensitizer based on experimental in situ UV-vis spectroscopy and resonance Raman spectro-electrochemistry, combined with quantum  chemical simulations performed at DFT levels of theory. 

The issue treated in this paper have relevance in photocatalysis and photovoltaics. I believe that the paper could be accepted for publication in this Journal after some minor revisions.

The abstract section must be improved by discussing mainly the major findings reported in the paper and avoiding sentences which represents the state of the art of the treated issue.

The authors have to better underline in the introduction section the aim and the novelty of their work with respect to what reported in literature by other authors.

Author Response

Revision of manuscript “Resonance Raman Spectro-Electrochemistry to Illuminate Photo-Induced Molecular Reaction Pathways”

Dear Reviewer 2,

herewith we would like to submit the revised version of the manuscript “Resonance Raman Spectro-Electrochemistry to Illuminate Photo-Induced Molecular Reaction Pathways” to the special issue "Raman Spectroscopy – A Swiss Army Knife".

Both referees recommended publication with minor revisions. We thank the referees for their critical comments, which we considered in revising the manuscript at hand. For the ease of comparison, we highlighted the major changes using red font in the revised manuscript.

Referee 2 wrote:

The abstract section must be improved by discussing mainly the major findings reported in the paper and avoiding sentences which represents the state of the art of the treated issue.

We thank the referee for this remark. We have changed the abstract according the suggestions and write:

Abstract: Electron transfer reactions play a key role for artificial solar energy conversion, however, the underlying reaction mechanisms and the interplay with the molecular structure are still poorly understood due to the complexity of the reaction pathways and ultrafast timescales. In order to investigate such light-induced reaction pathways, a new spectroscopic tool has been applied, which combines UV-vis and resonance Raman spectroscopy at multiple excitation wavelengths with electrochemistry in a thin-layer electrochemical cell to study [RuII(tbtpy)2]2+ (tbtpy = tri-tert-butyl-2,2':6',2"-terpyridine) as a model compound for the photo-activated electron donor in structurally related molecular and supramolecular assemblies. The new spectroscopic method substantiates previous suggestions regarding the reduction mechanism of this complex by localizing photo-electrons and identifying structural changes of metastable intermediates along the reaction cascade. This has been realized by monitoring selective enhancement of Raman-active vibrations associated with structural changes upon electronic absorption when tuning the excitation wavelength into new UV-vis absorption bands of intermediate structures. Additional interpretation of shifts in Raman band positions upon reduction with the help of quantum chemical calculations provides a consistent picture of the sequential reduction of the individual terpyridine ligands, i.e., the first reduction results in the monocation [(tbtpy)Ru(tbtpy)]+, while the second reduction generates [(tbtpy)Ru(tbtpy)]0 of triplet multiplicity. Therefore, the combination of this versatile spectro-electrochemical tool allows to deepen the fundamental understanding of light-induced charge transfer processes in more relevant and complex systems.

The authors have to better underline in the introduction section the aim and the novelty of their work with respect to what reported in literature by other authors.

We have changed the introduction section. We now write on page 2, line 50:

However, the characterization of intermediate states is challenging due to their high reactivity, still comparably short lifetimes and low concentrations. In this contribution a new setup utilizing a custom-made thin layer spectro-electrochemical cell is realized, which combines UV-vis- and resonance Raman (RR) spectro-electrochemistry (SEC) to study electronic and structural changes during one- and multielectron reduction.

We also highlight the novelty of the experimental results at the end of the introduction. We write on page 2, line 94:

Therefore, the present joint spectroscopic-theoretical work provides a comprehensive study of the sequential electrochemical reduction of the [Ru(tbtpy)2]2+ photosensitizer based on experimental in situ UV-vis and RR-SEC corroborated by quantum chemical simulations performed at the density functional (DFT) and time-depended DFT (TDDFT) levels of theory. This approach provides to the best of our knowledge first and convincing experimental evidence for the previously proposed reduction mechanism by direct spectroscopic characterization of the structure of the singly and doubly reduced form of [Ru(tbtpy)2]2+.

We thank both reviewers for their helpful comments.

With kind regards also on behalf of Prof. Benjamin Dietzek and our co-authors,

Dr. Linda Zedler